# Compressing multidimensional weather and climate data into neural networks

**Langwen Huang**
Department of Computer Science
ETH Zurich
langwen.huang@inf.ethz.ch

**Torsten Hoefler**
Department of Computer Science
ETH Zurich
torsten.hoefler@inf.ethz.ch

## Abstract

Weather and climate simulations produce petabytes of high-resolution data that are later analyzed by researchers in order to understand climate change or severe weather. We propose a new method of compressing this multidimensional weather and climate data: a coordinate-based neural network is trained to overfit the data, and the resulting parameters are taken as a compact representation of the original grid-based data. While compression ratios range from 300× to more than 3,000×, our method outperforms the state-of-the-art compressor SZ3 in terms of weighted RMSE, MAE. It can faithfully preserve important large scale atmosphere structures and does not introduce significant artifacts. When using the resulting neural network as a 790× compressed dataloader to train the WeatherBench forecasting model, its RMSE increases by less than 2%. The three orders of magnitude compression democratizes access to high-resolution climate data and enables numerous new research directions.

## 1 Introduction

Numerical weather and climate simulations can produce hundreds of terabytes to several petabytes of data (Kay et al., 2015; Hersbach et al., 2020) and such data are growing even bigger as higher resolution simulations are needed to tackle climate change and associated extreme weather (Schulthess et al., 2019; Schär et al., 2019). In fact, kilometer-scale climate data are expected to be one of, if not the largest, scientific datasets worldwide in the near future. Therefore it is valuable to compress those data such that ever-growing supercomputers can perform more detailed simulations while end users can have faster access to them.

Data produced by numerical weather and climate simulations contains geophysical variables such as geopotential, temperature, and wind speed. They are usually stored as multidimensional arrays where each element represents one variable evaluated at one point in a multidimensional grid spanning space and time. Most compression methods (Yeh et al., 2005; Lindstrom, 2014; Lalgudi et al., 2008; Liang et al., 2022; Ballester-Ripoll et al., 2018) use an auto-encoder approach that compresses blocks of data into compact representations that can later be decompressed back into the original format. This approach prohibits the flow of information between blocks. Thus larger blocks are required to achieve higher compression ratios, yet larger block sizes also lead to higher latency and lower bandwidth when only a subset of data is needed. Moreover, even with the largest possible block size, those methods cannot use all the information due to computation or memory limitations and cannot further improve compression ratio or accuracy.

We present a new lossy compression method for weather and climate data by taking an alternative view: we compress the data through training a neural network to surrogate the geophysical variable as a continuous function mapping from space and time coordinates to scalar numbers (Figure 1). The input horizontal coordinates are transformed into three-dimensional Cartesian coordinates on the unit sphere where the distance between two points is monotonically related to their geodesic distance, such that the periodic boundary conditions over the sphere are enforced strictly. The resulting Cartesian coordinates and other coordinates are then transformed into Fourier features before flowing into fully-connected feed forward layers so that the neural network can capture high-frequency signals (Tancik et al., 2020). After the neural network is trained, its weights are quantized from

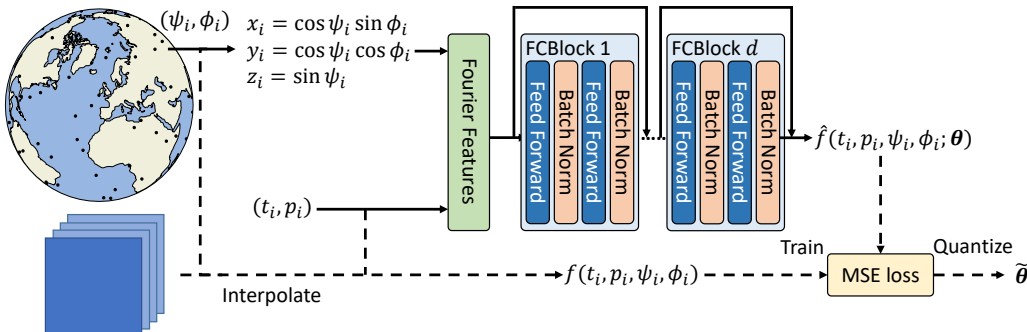

Figure 1: Diagram of the neural network structure: Coordinates are transformed into 3D coordinates and fed to the Fourier feature layer (green). Then they flow into a series of fully connected blocks (light blue) where each block consists of two feed forward layers (dark blue) with batch norms (orange) and a skip connection. Solid lines indicate flows of data in both compression (training) and decompression (inference) processes, and dash lines correspond to the compression-only process.

float32 into float16 to get an extra 2× gain in the compression ratio without losing much accuracy. This method allows compressing a large data block and squeezes redundant information globally while still having low latency and high bandwidth when retrieving decompressed values. It performs well in the high compression ratio regime: it can achieve 300× to more than 3,000× in compression ratio with better quality than the state-of-the-art SZ3 compressor (Liang et al., 2022) in terms of weighted root mean squared error (RMSE) and mean absolute error (MAE). The compression results preserve large-scale atmosphere structures that are important for weather and climate predictions and do not introduce artifacts, unlike SZ3. The neural network produced by our method can be used as an IO-efficient dataloader for training machine learning models such as convolutional neural networks (CNNs) and only lead to a limited increase in test errors compared with training with original data. Although equipped with Fourier features, error analysis shows that our method poorly captures rarely occurring sharp signals such as hurricanes: it tends to smooth out extreme values inside the center of a hurricane. It also suffers from slow training speed, especially in the low compression ratio regime.

Besides high compression ratios, our method provides extra desirable features for data analysis. Users can access any subset of the data with a cost only proportional to the size of the subset because accessing values with different coordinates is independent and thus trivially parallelizable on modern computation devices such as GPUs or multi-core CPUs. In addition, the functional nature of the neural network provides "free interpolation" when accessing coordinates that do not match existing grid points. Both features are impossible or impractical to implement on traditional compressors with a matching compression ratio.

## 1.1 RELATED WORK

**Compression methods for multidimensional data** Existing lossy compression methods compress multidimensional data by transforming it into a space with sparse representations after truncating, and then quantize and optionally entropy-encode the resulting sparse data. As one of the most successful methods in the area, SZ3 (Liang et al., 2022) finds sparse representations in the space of locally spanning splines. It provides an error-bounded compression method and can achieve a 400× compression ratio in our test dataset. TTHRESH (Ballester-Ripoll et al., 2018) compresses data by decomposing it into lower dimensional tensors. It performs well on isotropic data such as medical images and turbulence data with compression ratios of around 300×, but it is not the case for heavily stratified weather and climate data where it either fails to compress or results in poor compression ratios of around 3×. ZFP (Lindstrom, 2014) is another popular compression method that provides a fixed-bitrate mode by truncating blocks of orthogonally transformed data. It provides only low compression ratios that usually do not exceed 10×. While there are lossless methods for multidimensional data (Yeh et al., 2005), they cannot achieve compression ratios more than 2× because scientific data stored in multidimensional floating point arrays rarely has repeated bit patterns (Zhao et al., 2020). SimFS (Girolamo et al., 2019) is a special lossless compression method that compresses the simu-

Table 1: Comparison of methods for multidimensional compressing weather and climate data with the same RMSE bound. The decompression speed is measured in both continuous and random access patterns. SimFS' decompression speed depends heavily on the compression ratio.

| Method | Comp. ratio | Comp. speed | Decomp. speed | |
| --- | --- | --- | --- | --- |
| | | | Cont. | Rand. |
| SimFS (Girolamo et al., 2019) | Arbitrary | | | |
| ZFP (Lindstrom, 2014) | $< 10\times$ | | | |
| TTHRESH (Ballester-Ripoll et al., 2018) | $< 300\times$ | | | |
| SZ3 (Liang et al., 2022) | $< 400\times$ | | | |
| NN (Ours) | $300\times$ - $3,000\times$ | | | |

lation data by storing simulation checkpoints every N time steps and, whenever needed, recreating the discarded result by re-running the simulation starting from the nearest saved time step. It can reach arbitrary high compression ratios by simply discarding the respective amount of intermediate results, but reconstruction (decompression) may be expensive because it always produces all the prognostic variables of a simulation. Table 1 summarizes the performance of the methods above when compressing weather and climate data.

**Neural representation**   Using neural networks to represent multidimensional objects is gaining popularity after the "spectral bias" of the vanilla MLP is resolved by Fourier features (Tancik et al., 2020) or SIREN (Sitzmann et al., 2020). Previous works of applying neural representation for compression focus on compressing 3D scenes (Mildenhall et al., 2021), images (Dupont et al., 2021), videos (Chen et al., 2021), 3D volumetric data (Lu et al., 2021), and isotropic fluid data (Pan et al., 2022). Instead of optimizing the compression ratio, it is also possible to optimize training time and learned details by adding a trainable embedding layer (Müller et al., 2022; Barron et al., 2021). Our work differs because the target data is defined on a manifold instead of the Euclidean space. Also, weather and climate data are typically larger, smoother, and much more anisotropic. COIN++ (Dupont et al., 2022) and MSCN (Schwarz & Teh, 2022) applied meta-learning methods to compress time slices of climate data into latent vectors of several bits. The two methods can achieve an asympotic compression ratio of 3,000× assuming infinite number of time slices, yet the actual compression ratios are typically below 50× considering the size of neural network weights.

## 2 METHOD

A geophysical variable in weather and climate dataset can be seen as a scalar function of four inputs $f(t, p, \psi, \phi)$ with time $t$, pressure $p$, latitude $\psi$ and longitude $\phi$. Note that pressure is used in meteorology as the coordinate for vertical levels while the height of an air parcel with constant pressure becomes a variable. We propose using a neural network with weights $\boldsymbol{\theta}$ as a compact representation of the function $\hat{f}(t, p, \psi, \phi; \boldsymbol{\theta})$. As Figure 1 shows, the neural network is trained with $N$ random samples of the coordinates $\{(t_i, p_i, \psi_i, \phi_i) | i \in [1, N]\}$ as inputs, and evaluations of the function $\{f(t_i, p_i, \psi_i, \phi_i) | i \in [1, N]\}$ at those coordinates as targets, and use the mean squared error (MSE) as the loss function. In practice, given a weather and climate data stored in a four-dimensional array, one can implement the function $f(t, p, \psi, \phi)$ by returning the interpolated value at the input coordinate. Since the aim of the method is to "overfit" the neural network to the data, there is no need for generalization to unseen inputs. Therefore we use all the data available in the training process. After training, the weights $\boldsymbol{\theta}$ are quantized from float32 into float16 and then stored as the compressed representation of the original data. In the decompression process, the stored weights are cast back to float32 and loaded into the neural network along with coordinates of the data to produce decompressed values at those input coordinates. We remark that more complex quantization (Gholami et al., 2021) and sparsification (Hoefler et al., 2021) schemes could be used for further compressing of neural network parameters that we explain in the following.

## 2.1 Neural network structure

We use the fully connected MLP with skip connections as the backbone of the neural network (Figure 1). It contains a sequence of fully connected blocks (FCBlocks), and each FCBlock contains two fully connected feed forward layers with batch norms and a skip connection in the end. We use the Gaussian Error Linear Unit (GELU) (Hendrycks & Gimpel, 2016) as the activation function in feed forward layers to make the neural network a smooth function. The widths of feed forward layers $m$ and the number of FCBlocks $d$ are configurable hyperparameters (Table 2) that influence the compression ratio.

Table 2: Hyper-parameters

| Symbol | Explanation |
| --- | --- |
| LR | Learning rate |
| $c_t$ | Scaling factor for time |
| $c_p$ | Scaling factor for pressure |
| $m$ | Number of Fourier features |
| $\sigma$ | The standard deviation of Fourier features |
| $d$ | Number of FCBlocks |
| $w_i$ | Width of the i-th FCBlock |

Firstly, the input coordinates are normalized and then transformed into Fourier features. During the normalization process, time and pressure coordinates are normalized by two constants: $\hat{t} = t/c_t, \hat{p} = p/c_p$. Latitude and longitude coordinates are transformed into 3D Cartesian XYZ coordinates on the unit sphere: $x = \cos(\psi)\cos(\phi), y = \cos(\psi)\sin(\phi), z = \sin(\psi)$ such that two points close to each other on the sphere are also close to each other on the XYZ coordinates. This transformation can enforce the periodic property along longitude strictly. Then, the five coordinates $\boldsymbol{v} = [\hat{t}, \hat{p}, x, y, z]$ are transformed into Fourier features $\boldsymbol{\gamma}(\boldsymbol{v}) = [\cos(\boldsymbol{B}\boldsymbol{v}^T)^T, \sin(\boldsymbol{B}\boldsymbol{v}^T)^T]$ where cos and sin are applied element-wise. $\boldsymbol{B}$ is a blocked diagonal random matrix with $3m$ rows and 5 columns whose non-zero values are independently and randomly sampled from a Gaussian distribution with 0 mean and standard deviation $\sigma$. The $\boldsymbol{B}$ matrix is blocked in a way that there are only interactions among $x, y$, and $z$:

$$\boldsymbol{B}^{3m \times 5} = \begin{bmatrix} \boldsymbol{B}_t^{m \times 1} & & \\ & \boldsymbol{B}_p^{m \times 1} & \\ & & \boldsymbol{B}_{xyz}^{m \times 3} \end{bmatrix}.$$

The values of the matrix $\boldsymbol{B}$ are fixed for a neural network and setting it to trainable does not lead to a noticeable difference according to Tancik et al. (2020).

As weather and climate data is anisotropic, some geophysical variables may vary several magnitudes along the pressure and latitude dimensions. In that case, the neural network only captures variations along that dimension while ignoring variations along all other dimensions. Therefore, we scale the neural network output to appropriate dynamic ranges at that coordinate. The scaling function can be computed by interpolating mean and range along pressure and latitude dimensions.

## 2.2 Sampling method

As we train the neural network using a stochastic optimization method, it is important to feed the inputs randomly as required by the stochastic optimization method to have a good convergence rate (Bottou, 2004). To achieve that, we randomly sample coordinates for latitude, longitude, and time within the data definition region. The target values on those coordinates are then interpolated such that coordinates are not restricted to grid point coordinates of the data. This makes our method independent from the input data grid and helps the neural network learn the underlying smooth function from the data instead of remembering values at each grid point and outputting meaningless values anywhere else. We use a low discrepancy quasi-random generator (Owen, 1998) to reduce the occurrence of closely clustered coordinates as well as reduce regions that are not sampled in each batch which also helps the neural network learn uniformly over the sampling region. In addition, to avoid biased sampling over polar regions, the latitude $\psi$ and longitude $\phi$ are sampled according to the metrics on the sphere (Figure 1): $\psi = \frac{1}{2}\pi - \cos^{-1}(1 - 2y), \phi = 2\pi x$ where $(x, y)$ is drawn from the low discrepancy quasi-random generator within range $[0, 1] \times [0, 1]$. For pressure coordinates, we simply sample the exact grid points because they are usually very sparse with only tens of grid points, and interpolating on them introduces too much error.

## 3 EXPERIMENTAL EVALUATION

In this section, we test our method by varying model structure and data resolution, apply it to the training dataset of a CNN, and compare the resulting test errors. The data used for experiments are extracted from the ERA5 dataset (Hersbach et al., 2020) and interpolated to regular longitude-latitude grids. When computing the mean of errors such as RMSE and MAE over the longitude-latitude grid, we assign weights to each grid point that are proportional to the cosine of latitudes to balance the denser polar regions and the thinner equator region (see Appendix A.1). Throughout the experiments, we fix the number of FCBlocks $d$ to 12, the number of Fourier features $m$ to 128, and the standard deviation of Fourier features $\sigma$ to 1.6. We use the Adam optimizer (Kingma & Ba, 2014) with a learning rate of 3e-4. The neural networks are trained for 20 epochs where each has 1,046,437,920 randomly sampled coordinates (matching the total number of grid points in dataset 1) that are split into batches of 389,880 (matching the size of 3 horizontal slices in dataset 1) coordinates. All the experiments are performed on a single compute node with 3 NVIDIA RTX 3090 GPUs.

### 3.1 COMPRESSING DATA WITH DIFFERENT RESOLUTIONS

Table 3: Comparison of compressing data with different spatial and temporal resolutions.

| Dataset | 1 | 2 | 3 | 4 |
|---|---|---|---|---|
| Variable | | Geopotential | | |
| Time period | 2016 | 2016 | 1979-2018 | 1979-2018 |
| Spatial res. | 0.50° | 0.25° | 5.625° | 2.8125° |
| Temporal res. | 24h | 24h | 1h | 1h |
| Pressure levels | 11 | 11 | 1 | 1 |
| Data shape | (366,11,361,720) | (366,11,721,1440) | (350640,1,32,64) | (350640,1,64,128) |
| Data size | 3.9GB | 15.6GB | 2.7GB | 10.7GB |

We test the performance of our method by compressing geopotential data from the ERA5 dataset with various sizes and resolutions (Table 3). Geopotential is selected because it is the most important variable for weather and climate predictions (Rasp et al., 2020). We also compress the temperature data with the same formats with resulting error plots in Appendix A.4. With datasets 1 and 2, we test the performance for compressing 4D high spatial resolution data following the settings in Grönquist et al. (2021). Both datasets have 11 pressure levels which are meteorologically important (Ashkboos et al., 2022). The year of 2016 is selected arbitrarily. We expect a similar result when selecting a different year as shown in the error plots of compressing datasets with four different years (1998, 2004, 2010, 2016) in Appendix A.3. On the other hand, we test the performance for compressing high temporal resolution data on the 500hPa pressure level with datasets 3 and 4. Datasets 3 and 4 also can be used for training machine learning models for weather prediction under the Weather-Bench (Rasp et al., 2020) framework. Note that 0.25° is the highest spatial resolution and 1 hour is the highest temporal resolution available in the ERA5 dataset.

Table 4: Compression ratios of our method under different datasets and widths.

| Width | Model size | Dataset | | | |
|---|---|---|---|---|---|
| | | 1 | 2 | 3 | 4 |
| 512 | 13.86MB | 287.92 | 1,150.07 | 197.58 | 790.32 |
| 256 | 3.81MB | 1,046.91 | 4,181.83 | 718.43 | 2,873.73 |
| 128 | 1.15MB | 3,482.97 | 13,912.56 | 2,390.16 | 9,560.64 |
| 64 | 0.40MB | 9,913.17 | 39,597.75 | 6,802.84 | 27,211.37 |
| 32 | 0.18MB | 22,372.67 | 89,366.74 | 15,353.09 | 61,412.36 |

We use neural networks with different widths from 32 to 512 to create a spectrum of compression ratios (Table 4). Because larger models take more than 20 hours to train on our devices which are not practical , we only test our method on the high compression ratio regime (>200×). As a

comparison, we compress the same data using SZ3 (Liang et al., 2022) with max absolute error tolerances from 10 to 2,000 and resulting compression ratios ranging from 10× to 700×. Additionally, We tested ZFP (Lindstrom, 2014) and TTHRESH (Ballester-Ripoll et al., 2018) but they cannot reach compression ratios larger than 10× and thus are not included in the following analysis. For each compression result, we compute weighted RMSE, weighted MAE, max absolute error, and 0.99999-quantile of absolute error. The error is measured in the unit of $m^2/s^2$ (the same as geopotential). We also analyze error distribution and perform a case study for outputs on Oct 5th 2016 when hurricane Matthew emerged over the Caribbean Sea.

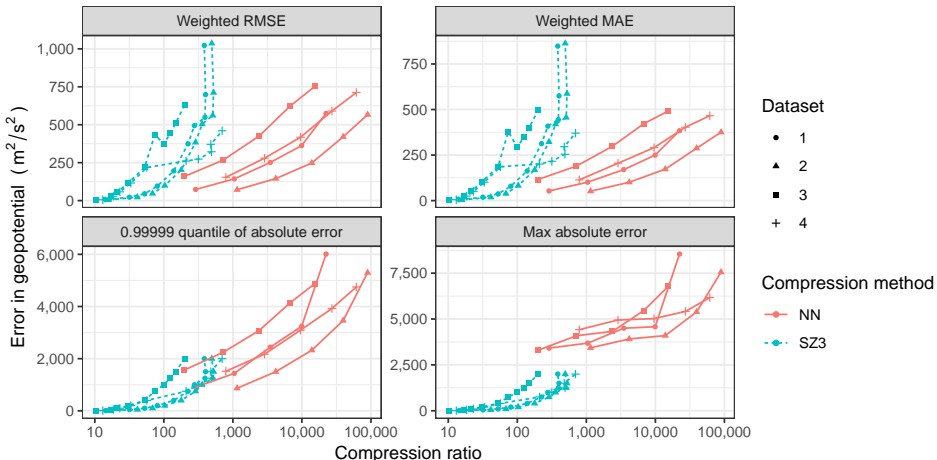

Figure 2: Weighted RMSE (upper left), weighted MAE (upper right), 0.99999-quantile of absolute error (lower left), and max absolute error (lower right) between the compressed and original data.

**Error analysis**    As shown in Figure 2, our method can achieve at least 10 times more compression ratios with the same weighted RMSE and MAE compared to SZ3. The gap between the two methods is even larger in higher error regions where our method can achieve more than 10,000× while SZ3 is capped at 600×. Comparing the results between datasets 1 and 2, 3 and 4 where the only spatial resolution differs, our method gets a free 4× gain in compression ratio without hurting any of the error measurements. This indicates our method can capture the real information underneath the continuous grid data. However, in terms of max absolute error, it is worse than SZ3 even with similar compression ratios due to different error distributions, as shown in Figure 3 (left). The majority of errors in our method concentrate near 0, but there exists a small fraction that is relatively large. That small fraction is around 0.001% according to the 0.99999-quantile of absolute error in Figure 2 where the two methods have similar errors at similar compression ratios. Meanwhile, SZ3 distributes errors more evenly within the error tolerance. In terms of the geographic distribution of errors, we calculate the mean and standard deviation of errors over time and pressure levels from compressed dataset 2 using our method (Figure 3 right). The mean values of errors of approximately -10 are distributed evenly over the globe with exceptions over mountainous areas. The standard deviation of errors shows meridional difference where it grows from the equator to polar regions.

**Case study**    Geopotential at 500hPa is one of the most important variables for weather prediction. Thus we select from Dataset 2 the 500hPa slice on Oct 5th 2016 when hurricane Matthew emerged over the Caribbean Sea and compare the compression results of the two methods with the original data. We use the biggest model with a compression ratio of 1,150× for our approach. For SZ3, we select the one with an absolute error tolerance of 1,000 and a compression ratio of 358× to match the 0.99999-quantile of absolute error.

In the global plot (Figure 4), our method clearly preserves the original data better despite having a much higher compression ratio: locations and shapes of minimum and maximum areas are nearly identical to the original ones, and the data is perfectly continuous over the $0°$ longitude line. The only noticeable difference is that some high-frequency details are smoothed out. On the SZ3 side, the average value over the equator region deviates from the original data, the structure of minimum

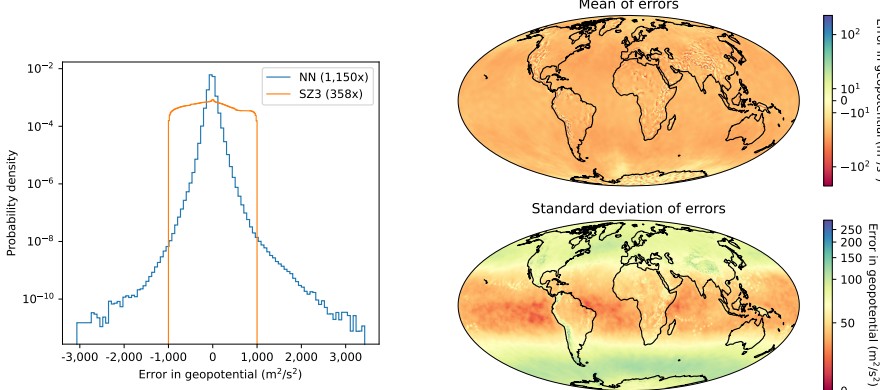

Figure 3: Left: Histogram of compression errors when compressing dataset 2 using our method with a width of 512 and SZ3 with an absolute error tolerance of 1000. Right: mean and standard deviation of compression errors of our method.

and maximum areas is broken due to numerous artifacts, and there is discontinuity crossing the $0°$ longitude line. Moreover, we can easily identify from data compressed by our method two subtropical highs over the North Atlantic and the North Pacific with slightly smoothed shapes compared to the original. The two subtropical highs are important for predictions of precipitations and tracks of tropical cyclones (hurricanes/typhoons). But from the SZ3 compressed data, we can only identify one over the Pacific while the other breaks into small pieces, which is meteorologically incorrect.

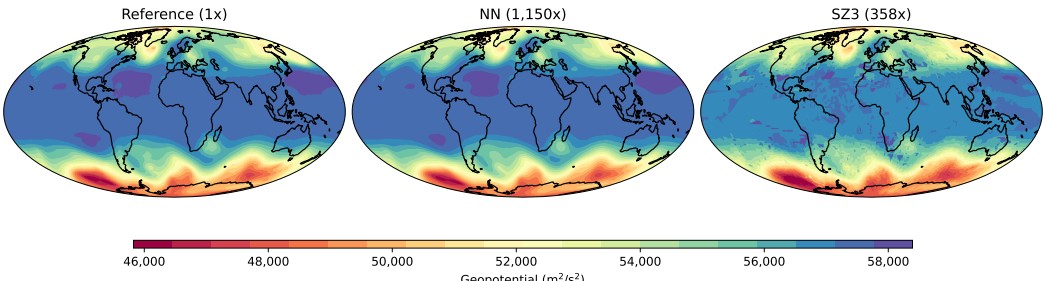

Figure 4: Global plots of geopotential at 500hPa on Oct 5th 2016 from original dataset 2 (left), compressed data using our method with a width of 512 (center), and SZ3 compressed data with an absolute error tolerance of 1,000 (right). See Appendix A.2 for plots with varying compression ratios.

In a closer look at the hurricane region (Figure 5), our method failed to capture the extreme values in the center of the hurricane although a smoothed version of the surrounding structure is preserved. SZ3 better preserves extreme value in the hurricane center but largely deviates from the original in other regions. As the error histogram in Figure 3 shows, the errors of our method in the center of the hurricane fall into the "tail" of the distribution that rarely occurs. Considering the hurricane center is just a few pixels wide, our method can provide a lot more details with fewer data compared with widely used lower resolution data that average out those extreme value regions.

## 3.2 COMPRESSING CNN TRAINING DATA

Training predictive machine learning models on large scientific datasets such as weather and climate data is often I/O bound (Dryden et al., 2021). Our method of compressing the dataset can potentially accelerate the training process by replacing the traditional dataloader with a trained neural network that generates training data on demand. Yet, it is unclear how point-wise error estimates such as RMSE or MAE of the training data relate to changes in test error of the trained neural network. Here, we apply our method to the input data of a CNN to show its effectiveness in this scenario.

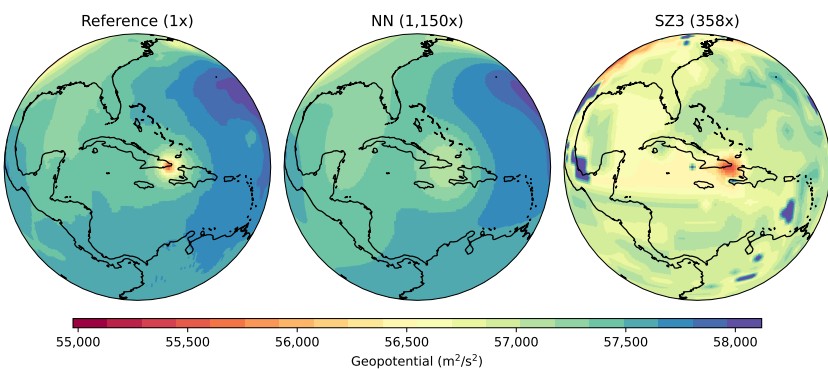

Figure 5: Local plots of geopotential at 500hPa on Oct 5th 2016 with hurricane Matthew in the center. The plots use data from original dataset 2 (left), compressed data using our method with a width of 512 (center), and SZ3 compressed data with an absolute error tolerance of 1,000 (right). See Appendix A.2 for plots with varying compression ratios.

We choose the WeatherBench (Rasp et al., 2020) baseline model as the CNN. We use geopotential at 500hPa in datasets 3 and 4 (Table 3) and temperature at 850hPa with the same range and shape. We select the biggest available neural network with a width of 512 to compress the geopotential and temperature data, respectively. For SZ3, we set the absolute error tolerances to 900 m$^2$/s$^2$ for geopotential and 5.0 °C for temperature to match the 0.99999-quantile of absolute error with our method. The outcomes are measured from the weighted RMSE error of CNN calculated using the test set.

Table 5: Weighted RMSE test error of CNN trained with original dataset and compressed dataset. C.R. represents compression ratio.

| Dataset 3 | C.R. | Weighted RMSE error (test set) | |
| --- | --- | --- | --- |
| | | Geopotential at 500 hPa (m$^2$/s$^2$) | Temperature at 850 hPa (°C) |
| Original | | 632.9 | 2.906 |
| NN compressed | 198× | 637.3 (+0.7%) | 2.944 (+1.3%) |
| SZ3 compressed | 71× | 650.6 (+2.8%) | 2.985 (+2.7%) |
| Dataset 4 | | | |
| Original | | 688.8 | 2.834 |
| NN compressed | 790× | 697.3 (+1.2%) | 2.888 (+1.9%) |
| SZ3 compressed | 106× | 702.9 (+2.0%) | 2.887 (+1.9%) |

**Results** The results in Table 5 shows that our method leads to small increases in weighted RMSE test errors of not more than 1.9% while achieving high compression ratios of 198× and 790× for dataset 3 and 4. On the other hand, SZ3 results in larger test errors and lower compression ratios of 71× and 106× probably due to the additional artifacts observed in Figure 4 and 5. The high compression ratio and limited increase in test errors of our method make it possible to feed machine learning models with much more data which will eventually surpass the model trained with original data in performance.

## 3.3 ABLATION STUDY

We perform an ablation study by selectively disabling some features of the model and comparing the resulting weighted RMSE loss between compressed data and original data. We select output scaling, XYZ transform, Fourier feature, skip connection, and batch norm as the targets because they are the special parts added to the fully connected MLP to compress data effectively. We also experiment changing the activation function from GELU to ReLU. We choose dataset 1 in Table 3 as the inputs and a neural network with a width of 512 which results in a compression ratio of 288×.

Table 6: Weighted RMSE between neural network compressed data and original data. In each experiment, parts of the model are disabled or replaced.Scaling means output scaling, XYZ means XYZ transform.

| | Scaling | GELU | XYZ | Fourier feature | Skip connection | Batch norm | WRMSE |
|---|---|---|---|---|---|---|---|
| 1 | | | | | | | 818 |
| 2 | ✓ | | | | | | 420 |
| 3 | ✓ | | ✓ | | | | 395 |
| 4 | ✓ | | ✓ | ✓ | | | 347 |
| 5 | ✓ | | | ✓ | | | 346 |
| 6 | ✓ | ✓ | | | | | 303 |
| 7 | ✓ | | ✓ | | ✓ | | 292 |
| 8 | ✓ | ✓ | ✓ | | ✓ | | 254 |
| 9 | ✓ | ✓ | ✓ | | ✓ | ✓ | 158 |
| 10 | ✓ | ✓ | ✓ | ✓ | ✓ | | 78 |
| 11 | ✓ | ✓ | ✓ | ✓ | ✓ | ✓ | 76 |

**Results**   As Table 6 shows, all of the features combined resulted in the lowest error. Output scaling, Fourier feature, skip connections and batch norm can indeed reduce the compression error compared with experiments without them. The GELU activation function can help reduce the error compared with ReLU when combined with output scaling and skip connection. The XYZ transform part does not have much impact on the final result, however, it is important to enforce the physically correct hard constraint on the boundary condition of output data.

## 4  LIMITATIONS AND FUTURE WORKS

Despite having high compression ratios and relatively low error, our method suffers from long compression/training time. It takes approximately 8 hours to train the network on dataset 2 (Table 3) with 4 NVIDIA RTX 3090 GPUs, whereas SZ3 takes just several minutes to compress with 32 CPU cores. The slow training process also limits us from using bigger models to get lower errors. Our method also struggles to capture small-scale extreme values as discussed in Section 3.1. While error analysis shows that it rarely occurs, such events can relate to very important weather processes like hurricanes.

We are going to investigate potential ways to integrate techniques like trainable embeddings (Müller et al., 2022; Barron et al., 2021) and meta-learning (Finn et al., 2017; Nichol & Schulman, 2018) to accelerate training and reduce extreme errors. In addition, we will expand our method to compress broader weather and climate data such as satellite images and weather station measurements into neural networks hoping to act as a complement to the traditional data assimilation in the numerical weather prediction pipeline (Bonavita et al., 2016).

## 5  CONCLUSION

We present a new method of compressing multidimensional weather and climate data into neural network weights. We test our method on subsets of the ERA5 dataset with different resolutions, and compare the results with the best available compressor SZ3 (Liang et al., 2022). Our method can achieve less weighted RMSE and MAE and similar 0.99999-quantile of absolute errors compared with SZ3 at compression ratios from 300× to 3,000×. While its max absolute error is worse than SZ3, the high errors are shown to be rare occurring (less than 0.001%). Through the case study on the hurricane Matthew event, our method can reproduce high-quality data that preserve the large-scale structure at a compression ratio of 1,150×, but it struggled to capture the extreme values in the hurricane center. It is also effective in replacing neural networking training data. Considering the compression ratio can be projected to 3,000× or even more on high-resolution data, our method has the potential to compress the whole ECMWF archive with hundreds of petabytes into tens of terabytes of data, fitting on a single hard drive, that can drive machine learning and other scientific applications.

**Reproducibility Statement**  We provide the source code implementing our method as well as instructions to reproduce the experiments in Section 3. The data we used is also publicly available through the download script in the source code. The source code is available in `https://github.com/huanglangwen/NNCompression`.

**Ethics Statement**  Our work can help reduce power consumption due to reduced data movement whenever loading the compressed data. It can make more weather and climate data easily available, which helps people better understand and mitigate climate change and severe weather.

**Acknowledgements**  This project received EuroHPC-JU funding under grant MAELSTROM, No. 955513. We thank the Swiss National Supercomputing Center (CSCS) for providing computing resources.

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

# A APPENDIX

## A.1 WEIGHTS FOR RMSE AND MAE ON THE REGULAR LONGITUDE-LATITUDE GRID

Both RMSE and MAE in this paper need to compute the average over the horizontal dimensions. The pointwise squared and absolute error can be seen as a function $f(\psi, \phi)$ mapping from latitude $\psi$, longitude $\phi$ to a scalar as the error that location. Ideally, the mean of the pointwise error over the sphere $\Omega$ is defined as:

$$\int_{\Omega} f(\psi, \phi)dS / \int_{\Omega} 1dS \tag{1}$$

. Both integrations can be decomposed into sums of integrations over individual cells $\Omega_{ij}$ in the regular longitude-latitude grid:

$$\sum_{i,j} \left( \int_{\Omega_{ij}} f(\psi, \phi)dS \right) / \sum_{i,j} S_{ij} \tag{2}$$

where $i$ denotes the cell index along the latitude dimension, $j$ denotes the index along the longitude dimension, $S_{ij}$ means the area of cell $\Omega_{ij}$. By approximating integrals using the product of cell center value and cell areas, we have:

$$\sum_{i,j} (f(\psi_i, \phi_j)S_{ij}) / \sum_{i,j} S_{ij} \tag{3}$$

. If the grid size is large enough (which is the case for datasets in Table 3), the cell area can be further simplified:

$$S_{ij} = r \cos(\psi_i) \Delta\psi \Delta\phi \tag{4}$$

where $r$ is the radius of the sphere, $\Delta\psi$ and $\Delta\phi$ are the height and length of each cell. Canceling out $r$, $\Delta\psi$ and $\Delta\phi$ as they are constant values in the regular longitude-latitude grid, the formula (1) becomes:

$$\sum_{i,j} f(\psi_i, \phi_j) \cos(\psi_i) / \sum_{i,j} \cos(\psi_i) \tag{5}$$

. It can be seen as the sum of pointwise errors $f(\psi_i, \phi_j)$ weighted by normalized cosine of latitudes $\cos(\psi_i) / \sum_i \cos(\psi_i)$.

## A.2 PLOTS OF GEOPOTENTIAL ON OCT 5TH 2016

In this section, we add extra plots to complement the case study in Section 3.1.

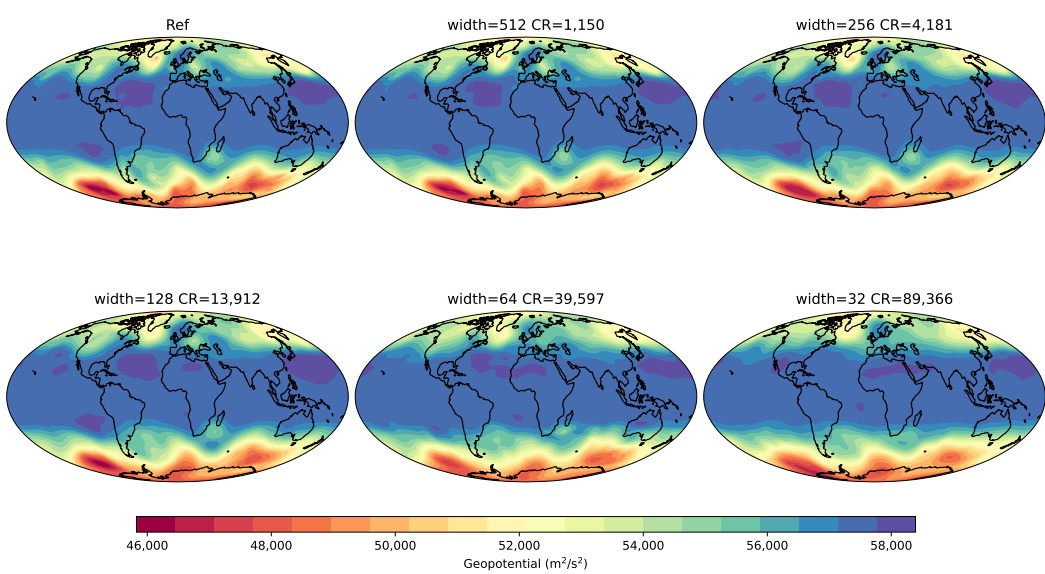

Figure 6: Global plots of geopotential at 500hPa on Oct 5th 2016 from compressed dataset 2 using our method with a range of widths. CR means compression ratio.

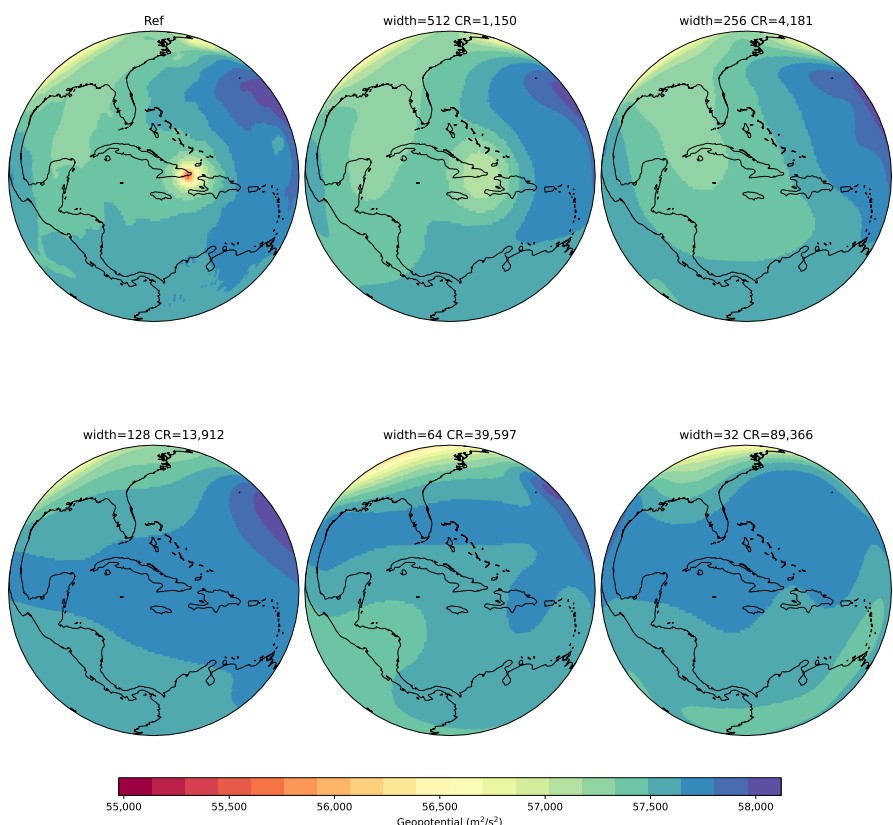

Figure 7: Local plots of geopotential at 500hPa on Oct 5th 2016 with hurricane Matthew in the center from compressed dataset 2 using our method with a range of widths. CR means compression ratio.

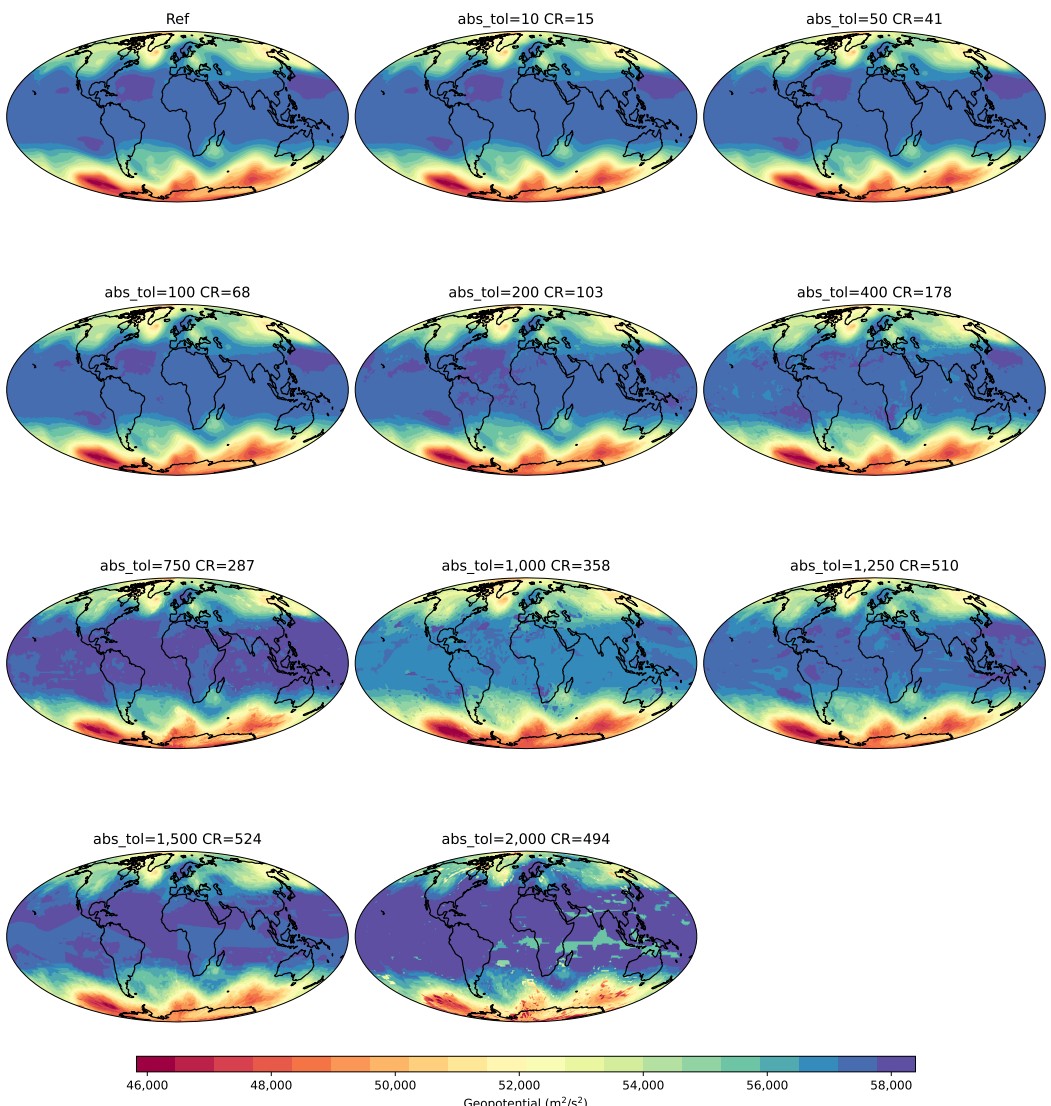

Figure 8: Global plots of geopotential at 500hPa on Oct 5th 2016 from SZ3 compressed dataset 2 with a range of absolute error tolerances. CR means compression ratio.

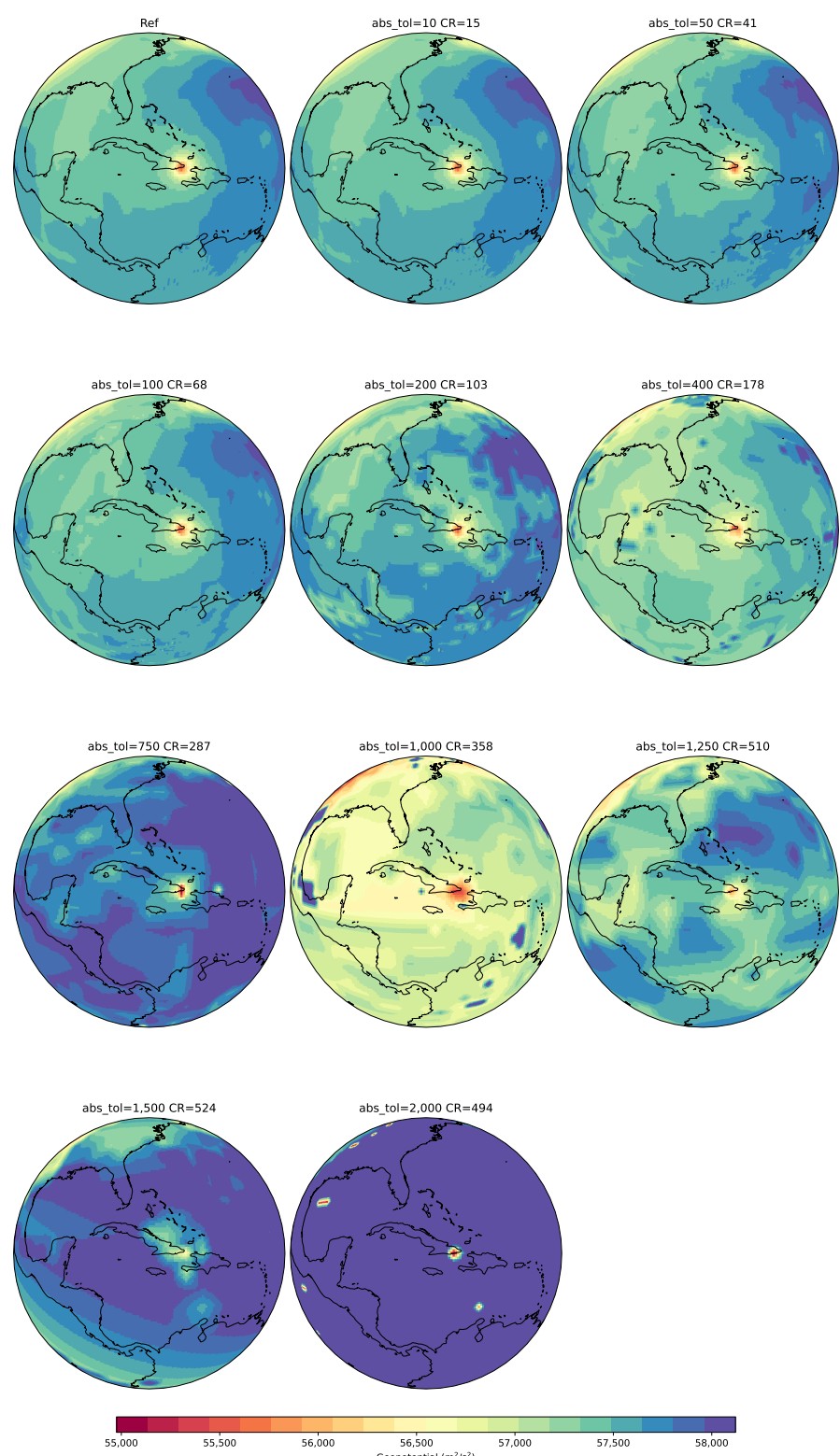

Figure 9: Local plots of geopotential at 500hPa on Oct 5th 2016 with hurricane Matthew in the center from SZ3 compressed dataset 2 with a range of absolute error tolerances. CR means compression ratio.

## A.3 Error plots for compression experiments with different years

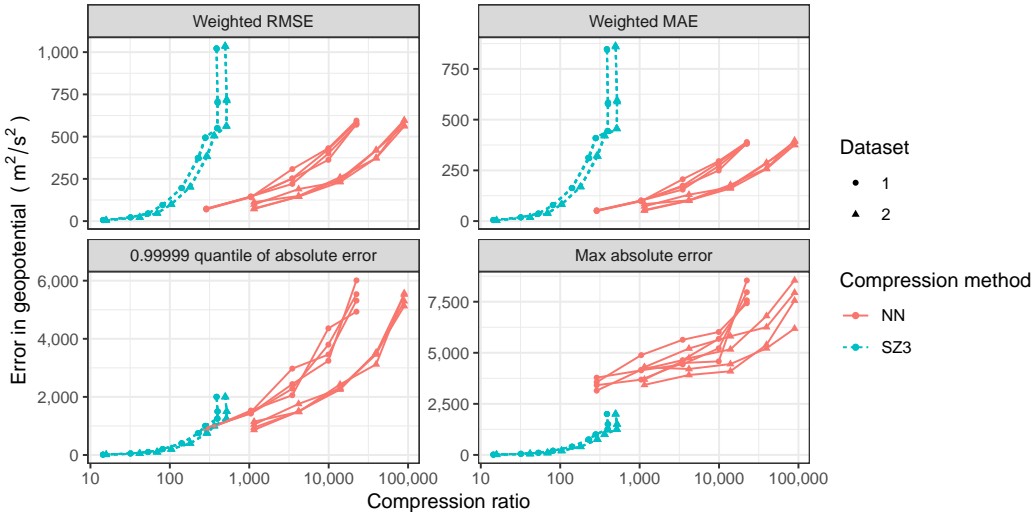

Figure 10: Weighted RMSE (upper left), weighted MAE (upper right), 0.99999-quantile of absolute error (lower left), and max absolute error (lower right) between the compressed and original geopotential data. Similar datasets with four different years (1998, 2004, 2010, 2016) are compressed.

## A.4 Error plots for compression experiments with temperature data

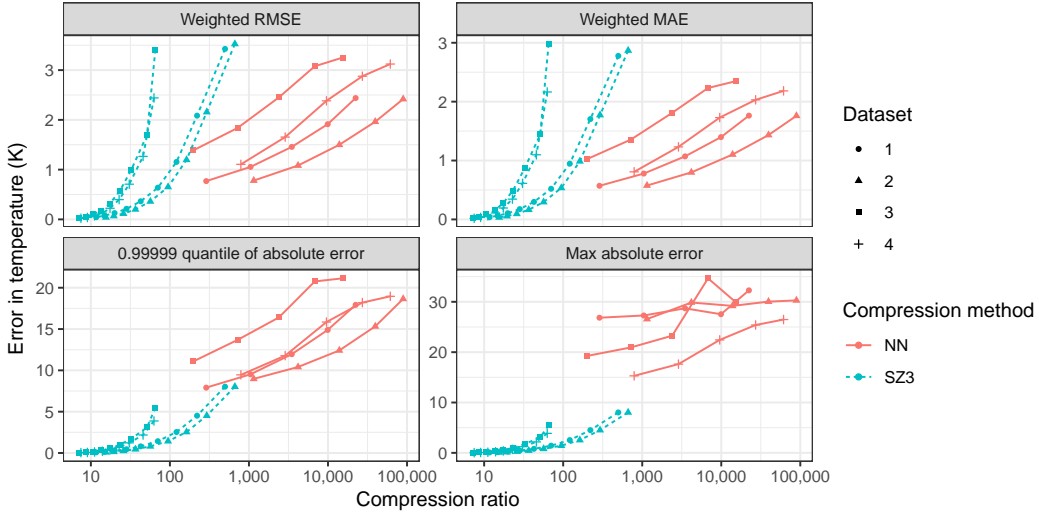

Figure 11: Weighted RMSE (upper left), weighted MAE (upper right), 0.99999-quantile of absolute error (lower left), and max absolute error (lower right) between the compressed and original temperature data.

