# OpenReview forum: "Compressing multidimensional weather and climate data into neural networks"
_ICLR.cc/2023/Conference — ICLR 2023 notable top 5%_

### Official Review · Reviewer_U9H2 · 2022-10-21

**Confidence:** 3
**Correctness:** 3
**Technical Novelty And Significance:** 3
**Empirical Novelty And Significance:** 3
**Recommendation:** 8

**Clarity, Quality, Novelty And Reproducibility:**

* The paper is well written and clear.
* The neural architecture used is based on Tancik (2006). There are some significant differences as the data points lie on the sphere rather than on a grid. I believe this to be of independent interest for the machine learning community.
* The experiments are ok (see above for more details).
* The code is provided, so reproducibility is good. I did not try running the code.

**Strength And Weaknesses:**

On the plus side,
* Weather data is an interesting application. The need for compression is also well motivated in the paper.
* I like the that the authors explain the specificities of weather data and how it impact their model.
* Compression is evaluated both in terms of reconstruction error and accuracy of a neural network trained on decompressed data. Results are good.

On the negative side,
* Experiments are a bit limited in scope. Only four data sets and one weather variable are considered. Unless I am mistaken, this means only four encoding were done. I appreciate that the data sets of different characteristics and I am aware encoding time is long. The largest time frame considered is 24 hours. The results are good, but they did not convince me the proposed approach could extended to years of historical data, an application mentioned by the authors. I think discussing this point would be interesting.
* Another application mentioned by the author is to compress simulation data. It would be interesting to compare compression time with simulation time.
* The use of GELU is not really motivated and not included in the ablation study.

-----------
**Update**
I had misunderstood the scope of the experiments. The authors pointed this out and were kind enough to add additional experiments. They also answered the other points I raised. I believe this work is worth publishing.


**Summary Of The Paper:**

This paper explores encoding high-resolution weather data using implicit neural representation (INR). In INR, a neural network takes as input a coordinate (here, transformed latitute, longitude, time and pressure) and output the compressed values (weather data here) at that coordinates. The approach is compared to several baselines and evaluated both in terms of compression error and in terms of performance of a model trained on the compressed data. The proposed approach significantly outperform the baselines. The paper also contains an ablation study.

**Summary Of The Review:**

For me this paper describes an interesting application, could have widespread use to compress weather data and some limited novelty on the machine learning side, so I find the content interesting. However, the limited scope of the experiments makes it impossible to know how it will behave on larger data sets such as all historical weather data.

---

> ### Author Response · Authors · 2022-11-14
> **Initial Response**
>
> > Experiments are a bit limited in scope. Only four data sets and one weather variable are considered. Unless I am mistaken, this means only four encoding were done. I appreciate that the data sets of different characteristics and I am aware encoding time is long.
>
> In section 3.2, we compressed the temperature data in addition to the geopotential data to train the CNN. We have also added experiments compressing temperature data in the same setting as in section 3.1. The error plots (in Appendix A.4) show similar results as compressing geopotential data.
>
> > The largest time frame considered is 24 hours.
>
> For dataset 1 & 2, they have 366 timesteps that cover the whole year of 2016, and the distance between two timesteps is 24h. For dataset 3 & 4, they have 350640 timesteps with one timestep every one hour spanning 40 years from 1979 to 2018.
>
> > The results are good, but they did not convince me the proposed approach could extended to years of historical data, an application mentioned by the authors.
>
> The year of 2016 in dataset 1 & 2 is selected arbitrarily. We have added experiments of compressing datasets in 1998, 2004, and 2010 (Appendix A.3), which all produced similar results. With this, we expect similar results for compressing datasets in any year.
>
> > Another application mentionned by the author is to compress simulation data. It would be interesting to compare compression time with simulation time.
>
> Typically, climate simulations have SYPD (simulation year per day) ranging from 1 to 36 [1], meaning that to produce simulation data of one year it takes 0.6 to 24 hours. Our method costs 8 hours to compress one-year data (similar to datasets 1 & 2) with the configuration of 512 in width and 12 in depth. However, the two run on vastly different platforms: climate simulations run on high-performance clusters with thousands of CPU cores, while our experiments run on a single server node with 4 NVIDIA RTX 3090 GPUs. Even if simulation time and compression time are unmatched (e.g. 0.6h vs 8h), we can pipeline the simulation and the compression process such that the compressed data can be produced with the same throughput as the uncompressed data.
>
> > The use of GELU is not really motivated and not included in the ablation study.
>
> We use GELU because it is a smoothed version of ReLU. It makes the overall neural network smoother and better in representing the smooth geophysical variable. We have added the GELU vs ReLU in the ablation study. The results show that using GELU can indeed reduce the compression error.
>
> ## Reference
>
> [1] Balaji, V., Maisonnave, E., Zadeh, N., Lawrence, B. N., Biercamp, J., Fladrich, U., ... & Wright, G. (2017). CPMIP: measurements of real computational performance of Earth system models in CMIP6. Geoscientific Model Development, 10(1), 19-34.

---

> > ### Comment · Reviewer_U9H2 · 2022-11-15
> > **Thank you**
> >
> > Dear authors,
> >
> > Thank you for pointing out my mistake about the datasets, for the new experiments and the comparison to simulation time.  I now believe that this paper does a good job showing the strength of the proposed approach and its practical usefulness.
> >
> > I think providing an xarray DataArray, as you proposed in another answer, would greatly facilitate adoption of your approach.
> >
> > I believe this paper is worth publishing in its current state. I will update my review accordingly.

---

### Official Review · Reviewer_5Wg4 · 2022-10-21

**Confidence:** 4
**Correctness:** 4
**Technical Novelty And Significance:** 3
**Empirical Novelty And Significance:** 3
**Recommendation:** 8

**Clarity, Quality, Novelty And Reproducibility:**

The paper is easy to read, the limits are clearly explained and the comparison with state-of-the-art also present.
The codes are present and seems to be easy to run.

**Strength And Weaknesses:**

Strengths:
- compression ratio (300 to 3000)
- spherical transformation, which is indeed very important for earth data
- usefulness of the method: while at first I was skeptical on the usefulness of such an approach for climate data, as clearly ERA5 will never be only stored in a degraded version, I was convinced by the usage of machine learning applications (and probably also others), where these amount of data are usually untractable for a local storage+usage.
- comparison with state-of-the-art and on different settings
- current limits of the method clearly analyzed

Some clarifications would be valuable for the paper, and in particular the following points:
- I think that in order to be used, this kind of approach would need to have an uncertainty quantification, at least based on the frequency of the signal to recover,  or based on the type of data, etc. in order for the user to be able to know when it can trust it and when in can't. How would you do it?
- The compression consists in training a neural network. Do you think this is robust enough to work with all size, input field, etc.? Do you think this could be used by a non-machine learning expert?
- In which type of applications would your error be negligible? It is good to have a very high compression, but the final quality needs to be satisfactory for targeted applications at least (since the hurricanes are for example not the case).
- The compressed data representation are the neural network weight values directly. Could your compression be used directly to feed a machine learning model, or would users have to decompress the data before using it?
- Would your method, and in particular the first part about the coordinate transformation, work in the case of non-worldwide data, i.e. only a part of the globe?


**Summary Of The Paper:**

This paper proposes a compression method for weather and climate worldwide data using a deep learning model combined with a Fourier transformation in order to enforce periodicity of the sphere data. The compression ratio is very high, from 300 to 3000, and the residual error comparable to methods with compression ratios drastically lower. However, the computation time is still high and high frequency features, such as hurricanes, are not always reconstructed. The tests were made on ERA5 reanalysis data, which is the open source widely used meteorological data set.

**Summary Of The Review:**

Though I am not a specialist in compressing data, I did use climate datasets for machine learning applications and I think this paper is worth publishing.

---

> ### Author Response · Authors · 2022-11-14
> **Initial Response**
>
> > I think that in order to be used, this kind of approach would need to have an uncertainty quantification, at least based on the frequency of the signal to recover, or based on the type of data, etc. in order for the user to be able to know when it can trust it and when in can't. How would you do it?
>
> This is a great point. Thank you! The resulting neural network can be shipped with measurements of posterior uncertainties such as quantiles of errors. The quantiles of errors can be calculated by evaluating the neural network at all grid positions of the original data, comparing with the original values, and reducing along the time dimension. The results can be stored as a lower dimensional array with dimensions of pressure, latitude, and longitude and (optionally) subsampled to a lower resolution such that it will not hurt the compression ratio.
>
> > Do you think this is robust enough to work with all size, input field, etc.?
>
> To achieve an optimal compression ratio, the input size cannot be too large that the capacity of the neural network cannot capture most of the information, and resolutions have to be high enough that most of the data points in the multidimensional array are redundant. If the data size is too large, we can split the data into blocks of smaller sizes and compress those blocks in parallel.
>
> Our method is designed for compressing smooth geophysical fields, which is the case for most weather and climate variables. It may encounter some difficulties when compressing variables that are not very smooth such as precipitation and cloud cover ratio.
>
> > Do you think this could be used by a non-machine learning expert?
>
> Yes, the trained neural network can be deployed as a standalone binary or Python package that runs on a wide range of devices. We can further develop a plugin for the xarray [1] package, so that users can access compressed data (without decompressing everything in memory) just like using the native xarray DataArray.
>
> > In which type of applications would your error be negligible? It is good to have a very high compression, but the final quality needs to be satisfactory for targeted applications at least (since the hurricanes are for example not the case).
>
> Other than ML applications as we have shown in section 3.2, our method can be used for climatology calculations where average values are calculated over the globe and several timesteps. This is much faster than calculating averages over the original multidimensional data because of the reduction in data movement. Moreover, current weather and climate simulations only save the states every several timesteps due to the limitations in storage. Our method can compress those intermediate simulation data while the compression errors can be ignored as the data would not be available otherwise.
>
> > Could your compression be used directly to feed a machine learning model, or would users have to decompress the data before using it?
>
> Our method can be used as a dataloader that generates slices of data on the fly when training a machine learning model. There’s no need to decompress the data before using it.
>
> > Would your method, and in particular the first part about the coordinate transformation, work in the case of non-worldwide data, i.e. only a part of the globe?
>
> Our method and the coordinate transformation work without problem for non-worldwide data. And there will be no loss in compression ratios because the capacity of the neural network does not depend on the range input coordinates. Moreover, the standard deviation of elements in $\mathbf{B}$ can be tuned such that $\mathbf{B}\hat{v}$ has a similar range with $\mathbf{B}v$ where $\hat{v}$ is the input coordinates limited to a small area, and $v$ is the input coordinates scattered across the globe.
>
> ## Reference
>
> [1] Hoyer, S., & Hamman, J. (2017). xarray: ND labeled arrays and datasets in Python. Journal of Open Research Software, 5(1).

---

### Official Review · Reviewer_Kcey · 2022-10-30

**Confidence:** 3
**Correctness:** 3
**Technical Novelty And Significance:** 2
**Empirical Novelty And Significance:** 3
**Recommendation:** 8

**Clarity, Quality, Novelty And Reproducibility:**

On a high level the paper is clear to follow and seems novel as detailed in the related work and methods section. There are a few implementation details which are unclear (listed above), which make this difficult to reproduce.

**Details Of Ethics Concerns:**

None that I can see, the authors have satisfactorily addressed this towards the end of the paper.

**Strength And Weaknesses:**

Strength:
* The empirical evaluation and ablation studies are reasonably well detailed, showcasing how this is applied in real world scenarios.
* The paper is well organized, motivated and details the related work.

Weakness:
* Some of the important details are quite unclear.
     *It's quite unclear on what the described "target" scalar value is. How is the ground truth information sourced from?
     * Why are the Fourier features used?
     * Perhaps, I'm missing an obvious detail in information theory. How is the precision recovered during decompression  when converting model weights from float16 to float32?
* Are the RMSE, RMAE metrics over grids the best choice of metrics? Is there a metric that compares blurriness of the decompressed output with the ground truth?
* The technical novelty of the introduced Neural Network itself isn't quite significant.




**Summary Of The Paper:**

The  authors proposes  their compression approach for multidimensional weather data leveraging Neural Networks. The authors argue that this could provide compression ratios from 300x to 3,000x which is quite important in complex high resolution data regime. They evaluate this on the ERA5 weather dataset benchmarked against the SZ3 baseline. The author's experiment on different network architectures and  provide a case study on a specific hurricane event.

**Summary Of The Review:**

Overall, the work proposes an reasonably novel technique in an important domain. The authors conduct sufficient empirical evaluation. However, there are details which are not described clearly along with the technical novelty of the Neural network model.

---

> ### Author Response · Authors · 2022-11-14
> **Initial Response**
>
> > It's quite unclear on what the described "target" scalar value is. How is the ground truth information sourced from?
>
> The target scalar value is the data at the input coordinate interpolated from the original multidimensional data. We have updated the text in the method section to clarify this.
>
> > Why are the Fourier features used?
>
> Fourier features are used for better capturing high-frequency signals in the original data [1]. Without Fourier features, the coordinate-based neural network tends to learn an over-smoothed representation of the original data due to the spectral bias [1], thus leading to a higher compression error (as shown in the ablation section).
>
> > How is the precision recovered during decompression when converting model weights from float16 to float32?
>
> When converting weights from fp16 to fp32, it is a direct cast: the fraction part and exponential part in fp16 are extended with more 0s to match the size of fp32. The set of representable numbers in fp16 is a subset of representable numbers in fp32. By casting from fp16 to fp32, the underlying number is unchanged. The only information loss in the quantization process is when we quantize weights from fp32 to fp16 by dropping extra bits in the fractional and the exponential part.
>
> > Are the RMSE, RMAE metrics over grids the best choice of metrics?
>
> RMSE, RMAE along with max absolute error are widely used for evaluating compression methods of scientific datasets [2-4]. Existing compression methods like SZ3 [5] and TTHRESH [6] use these metrics for quality control.
>
> > Is there a metric that compares blurriness of the decompressed output with the ground truth?
>
> There are “perceptual losses” [7] that, to some extent, measure the blurriness of an image compared to the ground truth. However, the perceptual closeness does not necessarily relate to the usefulness of compressed weather and climate data. For example, a difference in the brightness of images can be ignored, but a shift in the mean values is unwanted in weather and climate data.
>
> > There are a few implementation details which are unclear (listed above), which make this difficult to reproduce.
>
> We provide the source code that implements our method. The results mentioned in the paper should be easily reproducible. We are also happy to answer any questions if there’s any issues when reproducing the results.
>
> ## Reference
> [1] Tancik, M., Srinivasan, P., Mildenhall, B., Fridovich-Keil, S., Raghavan, N., Singhal, U., ... & Ng, R. (2020). Fourier features let networks learn high frequency functions in low dimensional domains. Advances in Neural Information Processing Systems, 33, 7537-7547.
>
> [2] Tao, D., Di, S., Guo, H., Chen, Z., & Cappello, F. (2019). Z-checker: A framework for assessing lossy compression of scientific data. The International Journal of High Performance Computing Applications, 33(2), 285-303.
>
> [3] Papa, A., Durner, R., Edinger, F., & Kellerer, W. (2019, June). SDRBench: A software-defined radio access network controller benchmark. In 2019 IEEE Conference on Network Softwarization (NetSoft) (pp. 36-41). IEEE.
>
> [4] Grosset, P., Biwer, C. M., Pulido, J., Mohan, A. T., Biswas, A., Patchett, J., ... & Ahrens, J. (2020, November). Foresight: analysis that matters for data reduction. In SC20: International Conference for High Performance Computing, Networking, Storage and Analysis (pp. 1-15). IEEE.
>
> [5] Liang, X., Zhao, K., Di, S., Li, S., Underwood, R., Gok, A. M., ... & Cappello, F. (2022). SZ3: A modular framework for composing prediction-based error-bounded lossy compressors. IEEE Transactions on Big Data.
>
> [6] Ballester-Ripoll, R., Lindstrom, P., & Pajarola, R. (2019). TTHRESH: Tensor compression for multidimensional visual data. IEEE transactions on visualization and computer graphics, 26(9), 2891-2903.
>
> [7] Zhang, R., Isola, P., Efros, A. A., Shechtman, E., & Wang, O. (2018). The unreasonable effectiveness of deep features as a perceptual metric. In Proceedings of the IEEE conference on computer vision and pattern recognition (pp. 586-595).

---

### Author Response · Authors · 2022-11-14
**General Response**

We thank the reviewers for their comments and suggestions. We have updated the paper with experiments on compressing temperature data and compressing geopotential data with three more years based on the feedback. We will respond to questions in individual replies.

---

### Decision · Program_Chairs · 2023-01-20

**Decision:**

Accept: notable-top-5%

**Justification For Why Not Higher Score:**

N/A

**Justification For Why Not Lower Score:**

This is an import paper that tackles an important problem (compressing weather and climate datasets for dissemination) and does a very good analysis of the representation and downstream errors. All reviewers agree on a score of 8.

**Metareview: Summary, Strengths And Weaknesses:**

This paper proposes to compress peta-byte scale high-resolution data from weather and climate simulation, such as the WeatherBench dataset, using the weights of a neural network overfitted on all available training data. The neural network models weather variables as multidimensional scalar functions of time, pressure level and lat/long, first converted to cartesian coordinates then represented as fourier coefficients, before being fed to a residual fully connected network. This compression achieves lossy compression with 300x to 3000x reduction of size and negligible  impact on the RMSE metric used for evaluating WeatherBench predictions, and also outperforms SZ3. The authors methodically compare the representation errors made by the neural network at various compress levels vs. SZ3 and note that the compression does not represent rare events (like hurricanes) well, and also evaluate the downstream impact of compression upon training error in a prediction model.

Reviewers praised the empirical evaluation and ablations (Kcey, U9H2), the clarity and reproducibility of the paper (Kcey), the very high compression rate (5Wg4, U9H2), usefulness for dissemination (5Wg4, U9H2), the use of Fourier spherical coordinates (5Wg4).

Weaknesses included the lack of novelty (Kcey), poor compression of rare events . Reviewer Kcey had some technical questions that were alleviated. Reviewer 5Wg4 had various questions about uncertainty estimation, applicability to non-globe data, need for decompression (the method can work as a data loader). Reviewer U9H2 had questions about the scope of the experiments, that were clarified.

Based on the 8, 8, 8 scores, this paper should be accepted.

**Note From Pc:**

if the above contains the word "oral" or "spotlight" please see: "oral" presentation means -> notable-top-5% and "spotlight" means -> notable-top-25%. As stated in our emails, we are disassociating presentation type from AC recommendations

**Summary Of Ac-Reviewer Meeting:**

N/A